# Seismic hazard analysis and financial impact assessment of railway infrastructure in the US West Coast: A machine learning approach

**Patcharaporn Maneerat[1], Panrawee Rungskunroch[1,2]*, Patricia Persaud[3]**

**1** Faculty of Engineering, Rajamangala University of Technology Thanyaburi, Pathum Thani, Thailand,
**2** Department of Industrial Engineering, Rajamangala University of Technology Thanyaburi, Pathum Thani, Thailand, **3** Department of Geosciences, University of Arizona, Tucson, AZ, United States of America

* panrawee.r@en.rmutt.ac.th

## Abstract

This research examines the seismic hazard impact on railway infrastructure along the U.S. West Coast (Washington, Oregon and California), using machine learning to explore how measures of seismic hazard such as fault density, earthquake frequency, and ground shaking relate to railway infrastructure accidents. By comparing linear and non-linear models, it finds non-linear approaches superior, particularly noting that higher fault densities and stronger peak ground shaking correlate with increased infrastructure accident rates. Shallow earthquakes with magnitudes of 3.5 or greater and hypocentral depths <20 km also exhibit a pronounced correlation with the incidence of railway infrastructure accidents The study extends to financial impact analysis through Net Present Value and Monte Carlo Simulation, and evaluates damage costs from 2000–2023 to guide financial planning and risk management strategies. It highlights the crucial role of advanced financial tools in optimizing maintenance and long-term planning that could result in better preparedness in high seismic hazard regions and emphasizes the need for robust risk management strategies in enhancing railway operational safety that considers the local and regional tectonic and seismic activity and local ground shaking intensity.

## 1. Introduction

### 1.1 Tectonic settings

The western United States is a region marked by intense tectonic activity, characterized by both aseismic and seismic phenomena that contribute to its dynamic landscape and topography. This area is defined by two primary tectonic boundaries. To the north, encompassing Washington, Oregon, and the northern edge of California, lies a subduction zone where the Juan de Fuca plate subducts beneath the North American plate. Conversely, the southern part, with predominantly California, features a transform plate boundary where the Pacific plate slides past the North American plate [1,2]. This transform boundary includes the San Andreas Fault, one of the world's longest strike-slip faults.

**Data Availability Statement:** All relevant data are within the manuscript and its Supporting Information files.

**Funding:** Rajamangala University of Technology's Postdoctoral and Postgraduate Talent Resource Development to encourage in deep research for improving competitiveness of industry contract code [B13F660068]. The funders had no role in study design, data collection and analysis, decision to publish, or preparation of the manuscript.

Beyond the San Andreas Fault, the western US is interlaced with numerous faults (Fig 1), which are sites of continuous tectonic activity [3]. This region has experienced a long history of earthquakes and aseismic deformation. Aseismic creep, a process where fault movement occurs without generating earthquakes, can be observed in several California locations, such as offsets at the University of California Berkeley's football stadium along the Hayward Fault, streets in Hollister near the Calaveras Fault, and in Parkfield on the San Andreas Fault [4–6]. Contrarily, seismic activity, or earthquakes, are notable for their capacity to cause significant ground motion variations that are challenging to predict with physics-based approaches [7] and extensive surface ruptures, as illustrated by the recent earthquakes and active faults (Fig 1), and major events such as the Ridgecrest earthquake in California [8].

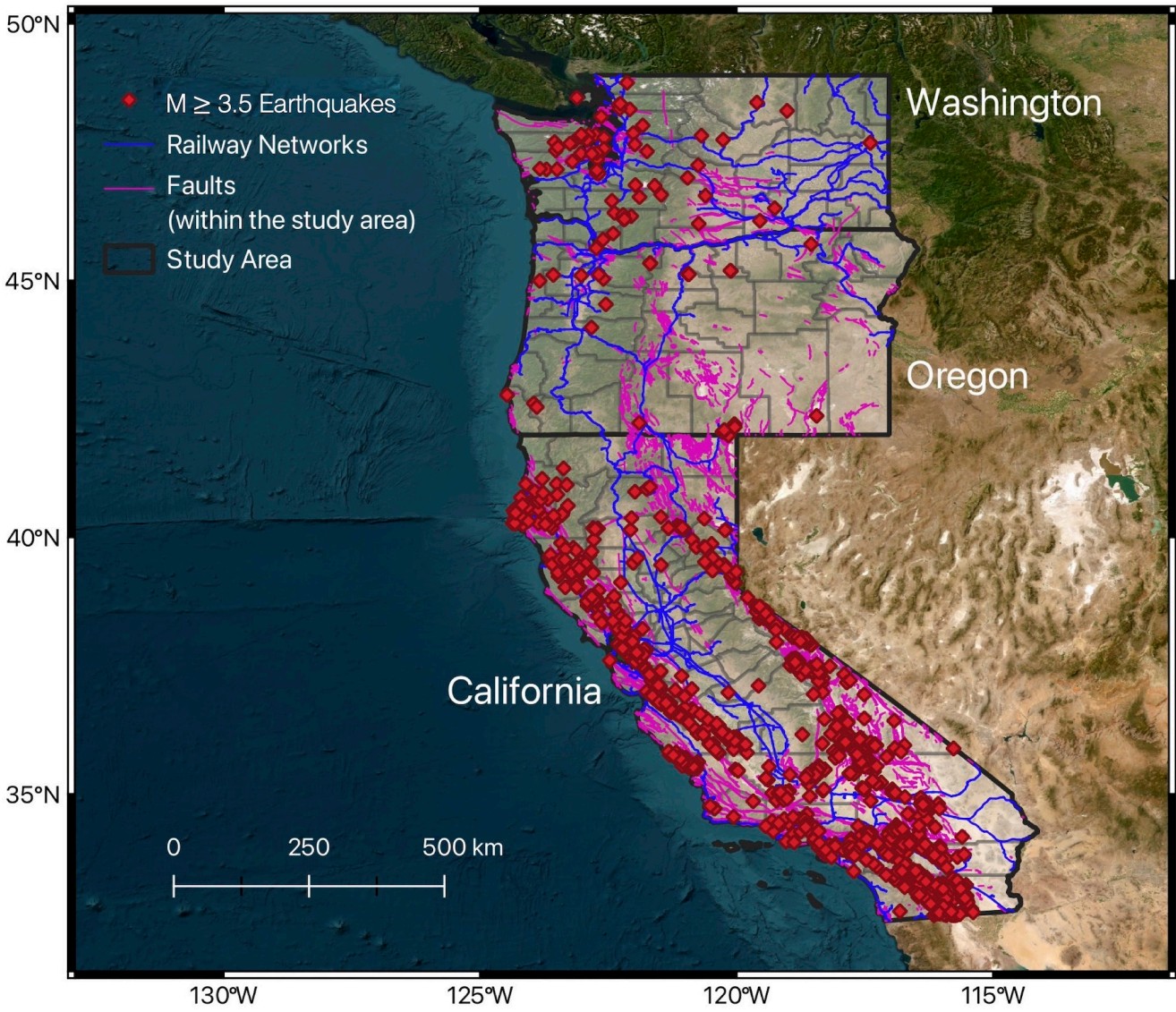

**Fig 1. Study region including the three states in the western U.S., Washington, Oregon and California.** The red diamonds represent magnitude ≥3.5 earthquakes occurring from 2000 to 2022 from the USGS catalog [9,10]. The pink lines indicate Quaternary faults [11] while the blue lines represent the rail networks [12]. The gray lines mark the counties within the study area [13]. The map was created using QGIS [14].

The impact of these tectonic movements varies, with aseismic creep and earthquakes affecting buildings and infrastructure in distinct ways. In areas less prone to frequent earthquakes, aseismic creep can still pose significant risks, as evidenced by the gradual displacement of UC Berkeley's football stadium, which extends a few millimeters annually. Earthquakes, however, have the potential to inflict more extensive damage. For instance, the 2022 magnitude 6.4 Ferndale earthquake, occurring approximately 10 miles southwest of Ferndale in Humboldt County, California, resulted in two fatalities, 17 injuries, and substantial damage to homes and infrastructure, including roads and utilities [15]. Understanding the diverse impacts of seismicity on infrastructure and buildings is essential for enhancing the resilience of our built environment and informing future urban and transportation planning efforts.

## 1.2 Historical impacts of earthquakes on rail networks in the Western US

Seismic events, along with their associated phenomena such as landslides and liquefaction, have historically posed significant threats to rail infrastructure, causing extensive damage to tracks, bridges, tunnels, and related facilities. This vulnerability is not unique to any single region, having been observed in various countries including Japan, Turkey, and the United States [16].

In the western United States, a number of seismic events have notably impacted rail networks. For instance, the 1999 magnitude 7.4 Hector Mine earthquake in California resulted in considerable destruction, damaging a stretch of rail track spanning 10 km [17]. Another poignant example is the 2001 magnitude 6.8 Nisqually earthquake in Washington, which caused extensive damage to bridges within a 55 km radius from the epicenter, with the seismic activity even reaching and affecting railroad buildings in Seattle [17].

Furthermore, earthquakes can also devastate other critical railway components such as tunnels, signaling, and communication facilities. The 1952 Kern County earthquake in California serves as a stark example, where tunnels intersected by secondary faults around 48 km from the epicenter sustained severe damage [16]. This event also led to buckled tracks and widespread infrastructure damage. Similarly, the Hector Mine and Nisqually earthquakes resulted in significant signal damage, affecting areas approximately 40 and 110 km away from the epicenters, respectively.

These instances illustrate how a single seismic event can produce damage over a broad area, affecting not only the tracks but also bridges, tunnels, and other critical infrastructure. The 1952 Kern County earthquake, for instance, not only damaged local rail facilities but also extended its impact to railway infrastructure in Santa Barbara County, California [16]. Given the complex and sometimes indirect relationship between seismic events and infrastructure damage, this study aims to explore the connection between them. It seeks to determine whether a simple linear model can adequately explain these relationships or if other factors play a significant role in railway infrastructure damages. The goal is to enhance our understanding of these dynamics to improve resilience and safety in rail network design, location choice and maintenance.

## 1.3 Financial assessment and sensitivity analysis

**1.3.1 An analysis of Net Present Value (NPV).** The NPV analysis is a crucial technique in evaluating the long-term profitability of a capital investment project. The analysis fundamentally considers the time value of money, emphasizing that the present value of a sum is not representative of its future worth. This concept, integral to investment decisions, acknowledges that money today is worth more than the same amount in the future due to its potential earning capacity. NPV calculates the difference between the present value of cash inflows and

outflows over various time periods, a method essential for assessing the financial viability of any project. It helps in understanding the actual cost of a project when considering the time value of money, thereby allowing for a more accurate comparison with other potential investments. The NPV analysis is pivotal in ensuring that the selected project maximizes potential returns, considering both the scale of the investment and the time frame over which returns are expected [18].

$$NPV = \sum\nolimits_{n=0}^{N} \frac{R_n}{(1+i)^n}$$

where N = Total number of time periods (unit: year), $R_n$ = Net cash flow at a single period, n = Time period (unit: year), i = interest rate at the time of calculation.

Following the highlighted significance of NPV analysis across various sectors, including transportation, many researchers contribute and extend the discourse into specialized methodologies and case studies, further validating the versatility and critical importance of NPV in railway project assessments. A study highlighted the application of NPV in evaluating the socio-economic benefits of the Heartland light rail project in Kansas City, USA, underlining the method's ability to encompass broader societal impacts such as reduced automobile usage [19]. There is also a benefit-cost analysis model incorporating NPV to assess the safety and performance benefits against the costs of track upgrades in North America, aiming to minimize train derailments and enhance travel times [20]. In the context of High-Speed Rail (HSR) projects in Hong Kong and Mainland China, some existing research demonstrated NPV's effectiveness in validating project viability pre-investment and facilitating comparisons with alternative investments [21]. Furthermore, Rahman et al. (2019) combined NPV with internal rate of return (IRR) analyses for evaluating light rail services in Jakarta, indicating substantial financial benefits for both the local government and passengers through minimized service costs [22].

Toporkova and Shylo (2018) delved into the analytical support of financial analysis at railway transport, emphasizing the necessity of tailored financial analysis methodologies that reflected the railway industry's unique financial relationships and operational dynamics [23]. This study systematically explored the critical role of financial analysis in informing management decisions within the railway sector, thereby underpinning the foundational aspects of NPV calculations. Bai et al. (2011) discussed the financial analysis of railway construction projects from a network perspective, underscoring the comprehensive evaluation of project costs and benefits [24]. This analysis illuminated the nuanced considerations necessary for a thorough NPV assessment, including construction investments, operating costs, and the tangible benefits of improved transport quality and reduced costs. Their work asserted the indispensability of network-based financial analysis in ensuring the economic feasibility of railway projects. Another research presented a compelling case study on the electrification of the Cairo-Alexandria railway line, applying cost-benefit analysis to assess its economic and financial viability. Their findings, revealing a modest internal rate of return but substantial economic benefits when broader impacts were considered, illustrated the critical importance of comprehensive NPV analysis in capturing the full spectrum of a project's potential value. However, the study revealed a significant pain point in obtaining the necessary data and information for comprehensive cost-benefit analysis, which was crucial for accurate NPV calculations. The difficulty in data collection could hinder the assessment of project viability, especially in regions with less developed information systems [25,26].

Also, Akhtar et al. (2017) on optimizing literature search systems underscored the importance of efficient information retrieval in supporting multi-domain research, including

transportation economics [27]. While not directly related to NPV calculations, their research highlighted the broader academic infrastructure supporting rigorous financial analyses in the railway sector and beyond. These studies collectively underscored the complex interplay of financial, operational, and technological factors that had to be navigated in the NPV analysis of railway projects, reinforcing the methodology's critical role in guiding strategic investment decisions.

**1.3.2 Sensitivity analysis.** A sensitivity analysis is necessary to assess unusual circumstances that may arise during operations, such as natural disasters, vandalism, and unforeseen damages. One tangible benefit of conducting such an analysis is its applicability to financial projections for new projects, enabling the allocation of reserve capital [18,28]. The ISO 14040 standard recommends utilizing Monte Carlo Simulation (MCS) in life cycle analyzes to mitigate uncertainties. This approach has been employed in railway research, particularly in Life Cycle Assessment (LCA), where it minimizes uncertainties related to the recycling process. The probability density function for the triangular distribution is defined as follows:

$$f(a) = \begin{cases} 0 & a < x \\ \dfrac{2(a-x)}{(y-x)(z-x)} & x \le a \le z \\ \dfrac{2(y-a)}{(y-x)(y-z)} & z \le a \le y \\ 0 & a > y \end{cases} \tag{1}$$

where x is a lower limit value, y is an upper limit value, and z is a mode value; x ≤ y ≤ z

## 2. Methods

### 2.1 Data resources and processing

There are two main categories of data used in the first part of this study which are railway infrastructure accidents and seismic hazard variables.

**2.1.1 Railway infrastructure accidents.** The railway infrastructure accidents obtained from the US Federal Railroad Administration (FRA) include all incidents occurring on rail tracks and/or structures. The FRA is the US Department of Transportation's agency that collects accident information annually per county. In this study, we obtained the accident data for three states (Washington, Oregon, and California) on the West Coast of the US, occurring from 2000 to 2022 [29] (Fig 2). Primarily, 15.2% of railway infrastructure accidents are of the T110 type, characterized by a wide gauge and defective or missing crossties. The T002 type (washout/rain/slide track) and the T207 type (detail fracture and shelling/head check) follow, representing 9.4% and 7.7% of accidents, respectively. Note that in some counties, no incident is recorded in the dataset, shown as transparent polygons in Fig 2.

**2.1.2 Seismic hazard variables.** We use three main datasets, which are earthquakes, active faults, and ground shaking (peak ground acceleration) as indicators of seismic hazard and refer to them as seismic hazard variables, noting that geological complexity and local site conditions are important aspects that are not considered in this study

(1) Earthquakes

We collected information on M≥3.5 earthquakes that occurred from 2000 to 2022 in the western U.S (see Supplementary Information for additional details on the dataset; Fig 1). This dataset is from the U.S. Geological Survey (USGS) website that lists earthquakes and includes information on hypocentral depths, magnitudes, and geographical locations of the earthquakes

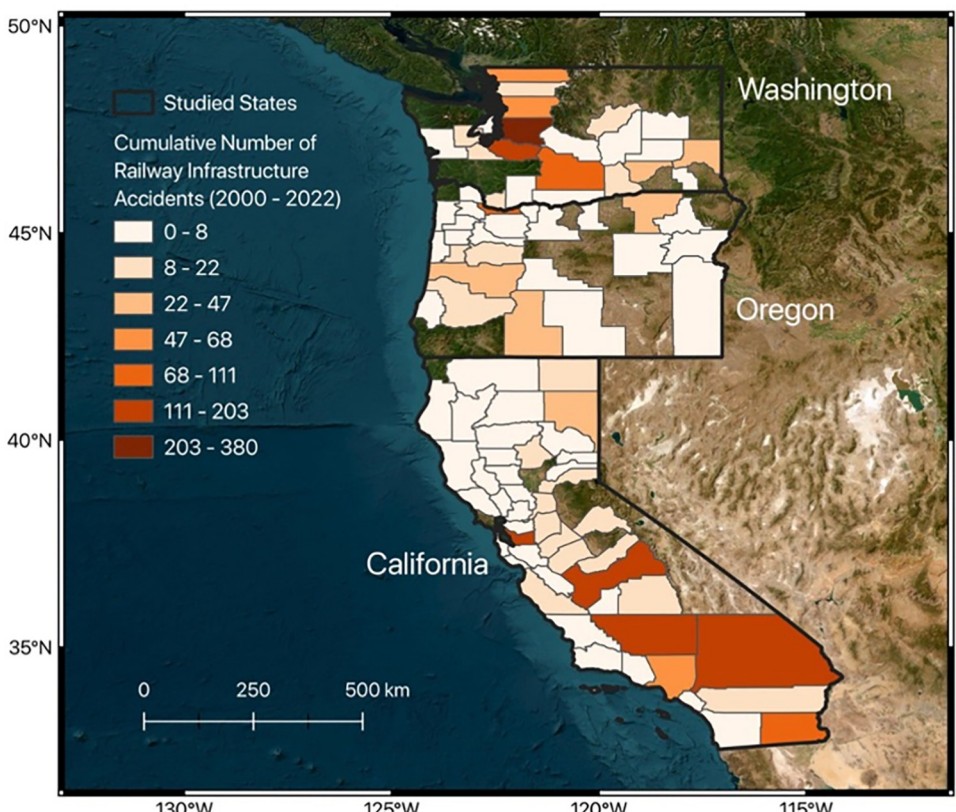

**Fig 2. Cumulative railway infrastructure accidents in each county occurring from 2000 to 2022 from the US Federal Railroad Administration.** The map was created using QGIS [14].

[9]. The total number of earthquakes in this dataset is 2,282 events. We selected earthquakes that had magnitudes greater than 3.5 because earthquakes starting from this magnitude can produce seismic waves that can be detected regionally by seismographs, offering more reliable data [30]. This facilitates the precise determination of earthquake parameters, including magnitude, location, and depth. Additionally, these earthquakes are more likely to cause noticeable surface effects, such as structural damage and felt ground shaking [31]. The maximum hypocentral depth among the events used in this study is 63 km occurring in Washington, and the highest magnitude among these earthquakes is M7.1 occurring in California. Since there is a wide range of both hypocentral depths and magnitudes of earthquakes (see Supplementary Information for additional details on the dataset), we divided the events based on these two factors. We then determined the number of earthquakes occurring annually in each county, using the following classification:

- EQ-MC1: Magnitude ≥M3.5 earthquakes, all depths,

- EQ-MC2: Magnitude ≥M4 earthquakes, all depths,

- EQ-MC3: Magnitude ≥M5 earthquakes, all depths,

- EQ-HD1: Magnitude≥M3.5 earthquakes and hypocentral depths ≤10 km,

- EQ-HD2: Magnitude ≥M3.5 earthquakes and hypocentral depths ≤20 km,

- EQ-HD3: Magnitude ≥M3.5 earthquakes and hypocentral depths ≤40 km.

(2) Active Faults

We acquired data on active faults in the western US from the US Geological Survey and California Geological Survey (2006) [11]. This dataset encompasses details such as fault names, types, lengths, and locations. Utilizing this information, we were able to calculate the fault density by determining the total length of faults per unit area within each county (Fig 3). This calculated metric was subsequently incorporated into our analysis to examine its significance as a predictor for railway infrastructure accidents. Areas in proximity to active fault lines are regarded as vulnerable zones, susceptible to damage from fault activity, whether through aseismic creep or earthquakes. By including fault density in our model, we aim to assess the long-term influence of fault activity on railway infrastructure, beyond just the immediate aftermath of seismic events. This approach allows us to gain a deeper understanding of the factors contributing to the resilience or vulnerability of railway systems in actively deforming regions.

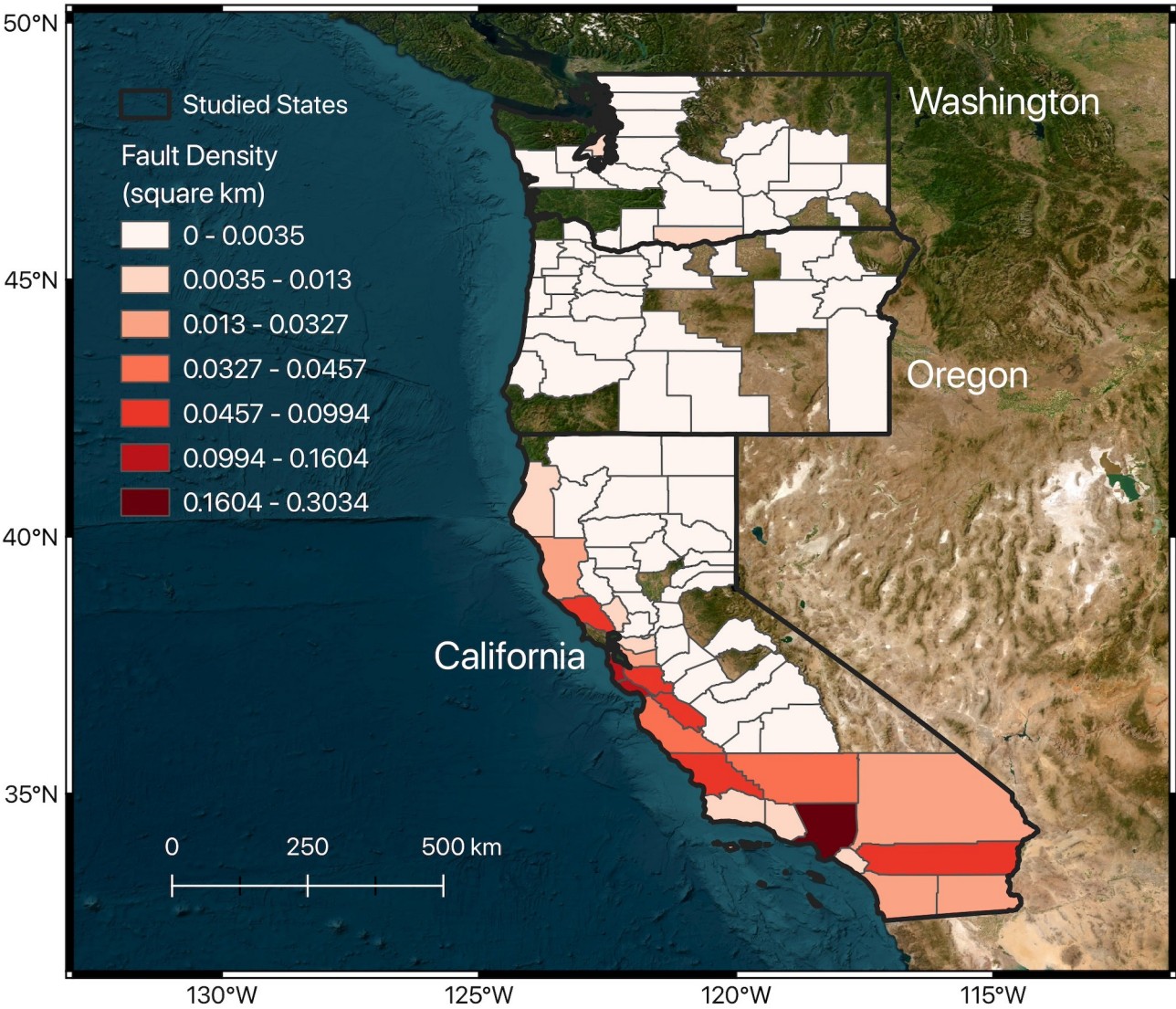

**Fig 3. Fault density map.** The density is calculated by dividing the length in km of the faults shown in Fig 1 by the size of each county in km². The map was created using QGIS [14].

(3) Peak Ground Acceleration (%g)

Peak Ground Acceleration (PGA) is a crucial indicator in seismology and earthquake engineering, denoting the maximum ground acceleration during an earthquake and measured in percentage of gravitational acceleration units (%g). It is critical for developing earthquake-resistant designs and assessing seismic risks. PGA at any location depends on factors such as earthquake magnitude (with larger earthquakes generating stronger motions), proximity to the epicenter, local geological features (soft sediments amplify shaking more than bedrock), earthquake depth (shallower earthquakes commonly produce higher PGAs), earthquake source characteristics, and the direction of the earthquake's rupture along with the site's topography [32]. These aspects collectively influence the experienced ground shaking, positioning PGA as essential for effective earthquake response and structural planning. The map depicting ground shaking in the western US was sourced from the National Atlas of the United States [33,34] (Fig 4) and has the grid spacing of 0.05 degrees for the conterminous United States. The map illustrates the likelihood of increases in peak ground motion to various levels, with Fig 4 specifically detailing the anticipated percentage of acceleration due to the gravity (%g). A higher PGA value in a region signifies a stronger peak horizontal ground acceleration, characterized by a 10% probability of exceedance within a 50-year period, compared to regions with lower PGA values. It's important to note that the hazard map we utilized from [33] is not the latest version available. However, we chose this map because it is more accessible and allows for easier extraction of the PGA values on a spatial basis. The most recent PGA map is available in the study conducted by [35].

In our research, we calculated the average ground shaking for each county and incorporated these averages as one of the predictive variables in our model. This approach allows us to quantitatively assess the potential impact of ground shaking on regional railway infrastructure by integrating a key measure of seismic intensity—peak ground acceleration—as a determinant of the vulnerability of railway systems to seismic events.

## 2.2 Modeling

In our research, we utilized two principal modeling approaches: linear regression and ensemble regression models. We chose the classical machine learning models primarily for their interpretability. Our aim is not only to predict outcomes but also to understand how individual features contribute to the model and their relationships with the target variables. The independent variables chosen to predict the number of accidents in each county include the annual cumulative number of earthquakes—as categorized in Section 2.1.2—alongside fault density and the average peak ground acceleration per county. It's important to note that our analysis does not incorporate other factors, such as the current condition of rail tracks, the last maintenance dates for tracks or structures, and the precise location and timing of accidents. This omission is due to the unavailability of such data. Additionally, Pearson's correlation between fault density and average PGA stands at 0.6, while the correlation between the number of earthquakes and average PGA is 0.4. These values reveal that the variables are not entirely independent, indicating a moderate degree of correlation. Nevertheless, this moderate correlation does not preclude their use in regression models. Through meticulous evaluation of the model and, if necessary, the implementation of regularization methods, it is possible to mitigate the impact of multicollinearity and maintain strong model performance. The modeling techniques applied in our study are detailed below.

**2.2.1 Linear regression.** Linear regression is a fundamental statistical technique that models the relationship between a dependent (target) variable and one or more independent (feature) variables by fitting a linear equation to observed data. It assumes a straight-line

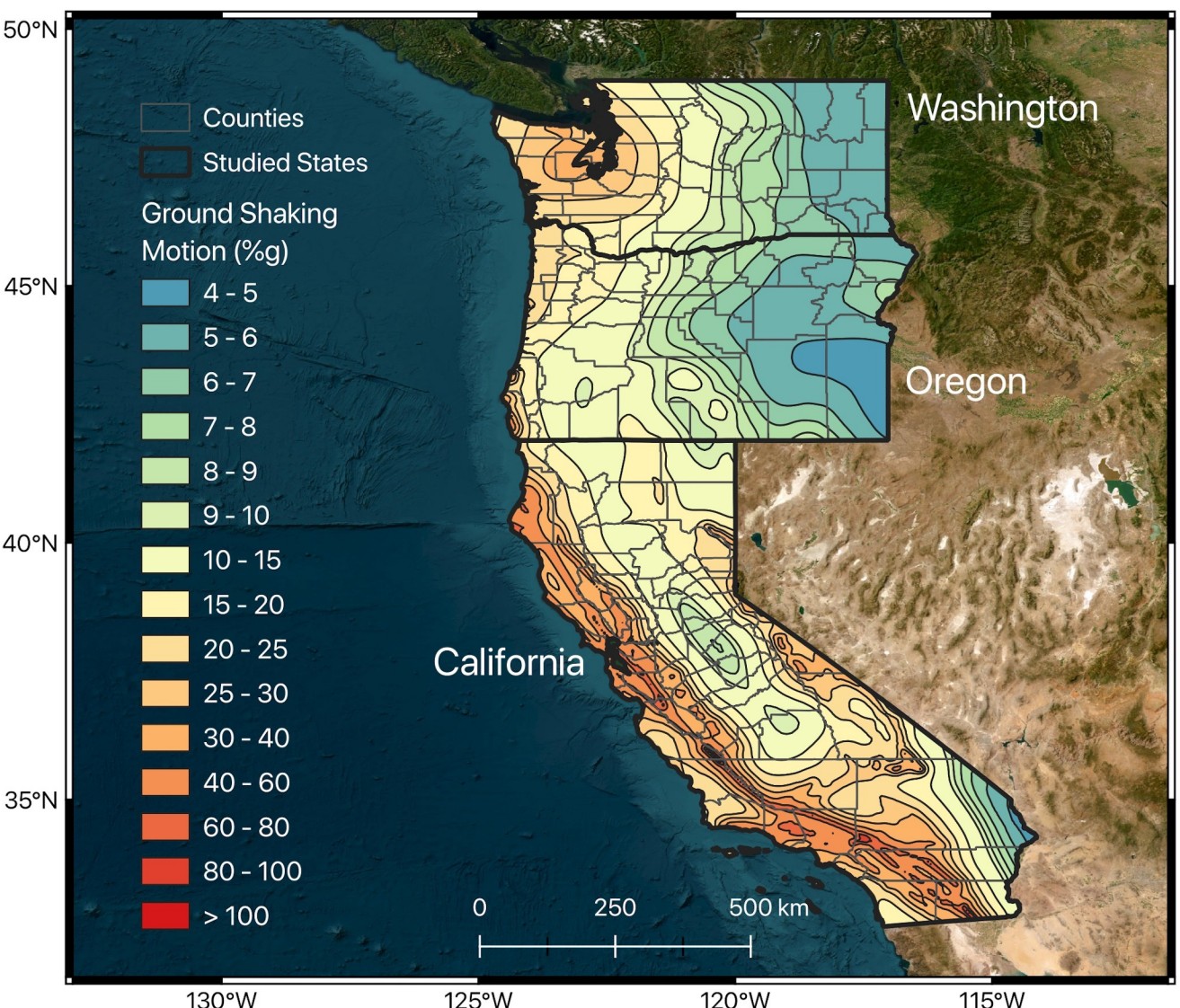

**Fig 4. Peak ground acceleration (%g) map as a measure of ground shaking in the western US obtained from [33,34].** The warmer colors represent higher peak horizontal ground accelerations with a 10% probability of exceedance in 50 years. The map was created using QGIS [14].

relationship whereby it attempts to predict the dependent variable's value based on the independent variables. The essence of linear regression is captured by the equation:

$$y_i = \beta x_i + \alpha \qquad (2)$$

where $y_i$ represents the dependent variable's value (target), $x_i$ denotes the independent variables (features), $\beta$ is the slope indicating the change in $y_i$ for a one-unit change in $x_i$, and $\alpha$ is the y-intercept, the value of $y_i$ when $x_i$ is zero.

This study explores four variations of linear regression models: simple linear (Least Squares), Lasso, Ridge, and Elastic Net regression. The latter three models incorporate regularization techniques to mitigate overfitting—a common issue where the model performs well on training data but poorly on unseen (test) data. Regularization works by imposing penalties on the size of the coefficients, effectively simplifying the model to improve its generalization

capabilities. Additionally, these regularization methods can address multicollinearity, a situation where independent variables are highly correlated, which can destabilize the standard least squares estimation. Unlike traditional least squares or simple linear regression, which solely focuses on minimizing the sum of squared differences between observed and predicted values, regularization adds a complexity penalty to the model selection criteria, balancing the trade-off between fit and complexity.

Lasso Regression, also known as Least Absolute Shrinkage and Selection Operator (L1 regularization), modifies traditional linear regression by introducing a penalty on the absolute size of regression coefficients. This technique encourages the reduction of less important feature weights to zero, thereby simplifying the model and aiding in feature selection. The goal of Lasso Regression is to minimize the loss function:

$$L1 = \sum_{i=1}^{n} (y_i - \alpha - \sum_{j=1}^{p} \beta_j x_{ij})^2 + \lambda \sum_{j=1}^{p} |\beta_j| \tag{3}$$

where the first term, $\sum_{i=1}^{n} (y_i - \alpha - \sum_{j=1}^{p} \beta_j x_{ij})^2$, represents the sum of squared residuals, and the second term, $\lambda \sum_{i=1}^{n} |\beta_j|$, is the penalty applied to the size of the coefficients, with $\lambda$ controlling the degree of weight shrinkage. As $\lambda$ increases, more coefficients are driven to zero, effectively eliminating less significant features from the model. This process of penalizing the sum of absolute values of the coefficients helps in dealing with multicollinearity and in automatically selecting the most relevant features for the model [36,37].

On the other hand, Ridge Regression, also known as L2 regularization, modifies linear regression by introducing a penalty on the square of the magnitude of coefficients. This technique aims to minimize the function [36,38,39]:

$$L2 = \sum_{i=1}^{n} (y_i - \alpha - \sum_{j=1}^{p} \beta_j x_{ij})^2 + \lambda \sum_{j=1}^{p} \beta_j^2 \tag{4}$$

where the first term, $\sum_{i=1}^{n} (y_i - \alpha - \sum_{j=1}^{p} \beta_j x_{ij})^2$, represents the sum of squared residuals, and the second term, $\lambda \sum_{j=1}^{p} \beta_j^2$, applies a penalty to the size of the coefficients squared, with $\lambda$ as the regularization parameter controlling the extent of the penalty applied. Unlike Lasso regression, which can completely eliminate coefficients, Ridge regression reduces the magnitude of coefficients but does not set them to zero. This method is particularly useful in preventing multicollinearity by ensuring that the model coefficients are not overly sensitive to changes in the model's input data, thereby reducing the model's complexity and improving its generalization capability [36,38,39].

Moreover, Elastic Net Regression effectively merges the attributes of L1 and L2 regularization techniques, integrating both absolute and squared value penalties to optimize the following objective function [40]:

$$L_{EN} = \frac{1}{2n} \sum_{i=1}^{n} (y_i - \alpha - \sum_{j}^{p} \beta_j x_{ij})^2 + \lambda \left[ \frac{1-\alpha}{2} \sum_{j=1}^{p} \beta_j^2 + \alpha \sum_{j=1}^{p} |\beta_j| \right] \tag{5}$$

where $\alpha$ is a parameter for the combination of Ridge ($\alpha = 0$) and Lasso ($\alpha = 1$). The Elastic Net uniquely blends these regularization methods, offering a tunable approach ($\lambda$ parameter) that linearly combines L1 regularization, associated with Lasso, and L2 regularization, identified with Ridge regression. This dual approach not only facilitates coefficient shrinkage but also enables the model to perform automatic feature selection by potentially reducing the coefficients of less significant variables to zero. Particularly advantageous in scenarios dealing with multicollinearity and overfitting, especially in high-dimensional datasets, Elastic Net stands out by leveraging the strengths of both Lasso and Ridge regression to produce a more robust

and versatile model [40]. This balanced mechanism makes Elastic Net highly effective for complex datasets where both the selection of relevant features and the mitigation of overfitting are crucial.

**2.2.2 Non-linear regression.** In this study, we employed Decision Tree and Random Forest regressions, both of which are non-parametric, supervised learning models designed for predicting target values. Each model employs a tree structure to facilitate regression analysis.

The Decision Tree Regression model constructs a tree from the top down, where each leaf represents a numerical prediction of the target value [41,42]. The root of the tree is selected based on the predictor (feature and threshold, e.g., Feature A < threshold) that minimizes the sum of squared residuals (SSR), where residuals are the differences between actual and predicted values. The tree grows by branching out from the root node, with each subsequent division designed to further reduce the SSR. The branching continues until terminal leaves, which represent the final predictions, are reached. While Decision Trees can adeptly handle complex datasets, they are prone to overfitting, particularly if the tree is allowed to grow too deep without restrictions. To mitigate this, we limited the maximum depth of the tree in our analysis, enhancing the model's generalizability to new, unseen data.

The Random Forest Regression model extends the concept of Decision Trees by constructing an ensemble of multiple trees [43,44]. It operates by randomly sampling observations and building several decision trees from these subsets. The final prediction is then derived by averaging the predictions from all trees. This methodology significantly reduces the risk of overfitting, a common drawback of single Decision Trees, by aggregating the diverse outcomes of multiple trees. The Random Forest model, by virtue of its ensemble approach, typically delivers more robust and accurate predictions than a single Decision Tree, making it a powerful tool for handling complex predictive tasks.

**2.2.3 Modeling.** In our study, we employed the scikit-learn library [45], a comprehensive tool within the Python programming language, for our modeling efforts. The modeling process was structured into three main stages: data preparation, hyperparameter tuning, and model execution.

We assumed that the residuals of the models follow a normal distribution, validated by examining the residual plots and conducting normality tests. It was also assumed that the observations are independent of each other, a crucial assumption for the validity of the statistical tests used. All data were standardized to have a mean of zero and a standard deviation of one, and missing values were imputed using the mean imputation method.

Initially, we divided our dataset, which includes all seismic variables and instances of railway infrastructure accidents as detailed in the Methods 2.1 section, into two distinct sets: training and test. The dataset was randomly split into training (80%) and testing (20%) sets to ensure that model performance was evaluated on unseen data. This randomization was performed using a fixed random seed to ensure reproducibility. We used 5-fold cross-validation during the training phase to optimize hyperparameters and prevent overfitting. The choice of 5-fold cross-validation was based on balancing computational efficiency with robust performance evaluation.

Following the data segmentation, we normalized our features to ensure uniformity in scale for both the training and test sets. Given that all our variables are numerical, we employed the MinMaxScaler function from scikit-learn, which adjusts the data to a range between 0 and 1 [45].

Subsequent to the scaling process, we embarked on hyperparameter tuning for our chosen models—Ridge, Lasso, Elastic Net Regression, and Ensemble Models—utilizing the GridSearch with 5-fold cross-validation from the scikit-learn library [45]. This step is crucial for identifying the optimal hyperparameters for each model, thereby enhancing their performance

**Table 1. Hyperparameters used in the GridSearchCV tuning techniques to find optimal values for each parameter of each model.**

| Ensemble Model | Hyperparameters [a] | Optimal Parameter Values Suggested by *GridSearchCV* |
|---|---|---|
| Decision Tree | max_depth | 6 |
| | min_samples_leaf | 6 |
| | max_features | log2 |
| | splitter | 'best' |
| Random Forest | max_depth | 9 |
| | min_samples_leaf | 6 |
| | max_features | 'auto' |
| | n_estimators | 500 |

[a] The hyperparameters are explained in [45].

[46]. For linear regression models, we focused on adjusting the alpha parameter ($\lambda$), as outlined in the Methods 2.2 section. For ensemble models, the hyperparameters tuned included splitter, max_depth, min_samples_leaf, and max_features for Decision Trees, and max_depth, max_features, min_samples_leaf, and n_estimators for Random Forests (Table 1).

Upon fine-tuning the hyperparameters, we proceeded with the model execution phase, resulting in the derivation of coefficient values for the linear regression models and feature importance for the ensemble models. To assess the accuracy of our models, we utilized the root mean square error (RMSE) metric, which will be elaborated upon in the subsequent section of our analysis.

**2.2.4 Modeling result validation metric.** As previously outlined, we employ the Root Mean Square Error (RMSE) metric to evaluate the performance of our models and to determine their efficacy in predicting the target variable—the number of railway infrastructure accidents. The RMSE serves as a measure for quantifying the prediction errors, essentially representing the average discrepancy between the observed actual values and the values predicted by our models. This metric is particularly useful for assessing the accuracy of predictions, as it provides insight into the average magnitude of prediction errors across all observations. The RMSE is calculated with the following formula [47,48]:

$$RMSE = \sqrt{\frac{1}{n}\sum_{i=1}^{n}|y_i - \hat{y}_i|^2} \tag{6}$$

where $y_i$ represent actual values or observations, $\hat{y}_i$ are predicted values, $n$ is a number of samples.

We chose RMSE as our model validation metric because it is particularly sensitive to large errors, squaring the deviations before averaging them. This sensitivity makes RMSE a stringent metric, ensuring that models with significant errors are appropriately penalized [47]. Given our study's focus on accurately predicting extreme values, such as large-scale railroad accidents due to major earthquakes, RMSE is essential for highlighting these deviations.

Additionally, RMSE provides a comprehensive measure of prediction accuracy by considering the magnitude of errors, offering a clear indication of model performance across all data points [47]. Expressed in the same units as the dependent variable, RMSE simplifies interpretation, helping us understand the scale of prediction errors in the context of railway accidents and infrastructure costs.

While other metrics, such as Mean Absolute Error (MAE) and R-squared are valuable, we opted for RMSE due to its robust evaluation capabilities and its ability to handle the complexity of our models. RMSE's sensitivity to larger errors provides a nuanced view of model

performance, especially in scenarios where significant errors can have substantial real-world implications.

To compare the performance of various models, we calculated the percentage differences in RMSE values between each model and the baseline Least-Squares (linear regression) model. This allowed us to determine whether alternative models offer significantly improved predictive accuracy for the target values. Additionally, we compared the average RMSE of linear regression models with regularization techniques (Ridge, Lasso, and Elastic Net) against the RMSE of non-linear models (decision trees and random forests). This comparison aimed to investigate whether more sophisticated modeling approaches can more accurately capture the relationship between seismic hazard variables and railway infrastructure accidents, thereby offering a deeper understanding of the underlying patterns and factors at play.

## 3. Modeling results

Upon executing the models, our analysis revealed that the linear model exhibits the highest RMSE, indicating its inferior performance compared to the alternative models (Table 2). While the Ridge, Lasso, and Elastic Net models outperformed the simple linear model, their RMSE values were still surpassed by those of the non-linear regression models. Notably, the Random Forest model achieved the most favorable outcome, reducing the RMSE by 62% relative to the simple linear model. Although the performance of the Decision Tree model did not quite match that of the Random Forest, the difference in their RMSE values was not markedly significant. Additionally, the normal distribution of residuals from the Decision Tree model suggests it effectively captures the principal patterns and variations within the data (Fig 5). Consequently, the insights derived from the Decision Tree model offer valuable explanations for the relationship between railway infrastructure accidents and seismic variables.

Further statistical analysis confirms the significance of these findings (Table 3). Paired t-tests between the linear model and each alternative model revealed extremely low p-values (all well below 0.05), indicating that the differences in RMSE values are statistically significant. For example, the t-statistic for the comparison between the linear model and the Random Forest model was 31.81, with a p-value of approximately 2.10e-38. These results underscore the superior performance of the non-linear models. Moreover, the 95% confidence intervals for the RMSE values of the alternative models (e.g., 5.66 to 6.19 for the Random Forest) further demonstrate the precision and reliability of these models. This comprehensive evaluation highlights the robustness of our findings and the effectiveness of using RMSE for model comparison, affirming that more sophisticated models like Random Forests provide significantly improved predictive accuracy over the simple linear model.

**Table 2. Modeling results.**

| Model | RMSE | % RMSE difference from Linear | % RMSE difference from Avg. Linear (with Regularization) |
|---|---|---|---|
| Linear | 15.538 | - | N/A |
| Ridge | 7.739 | 50% | - |
| Lasso | 7.742 | 50% | - |
| Elastic Net | 7.74 | 50% | - |
| Decision Tree | 6.167 | 60% | 20% |
| Random Forest | 5.926 | 62% | 23% |

[a] The table includes RMSE values and % differences between RMSE of other models and the simple linear regression model.

[b] The last column indicates the % differences between average RMSE of linear regression with regularization (Ridge, Lasso and Elastic Net) and each ensemble model.

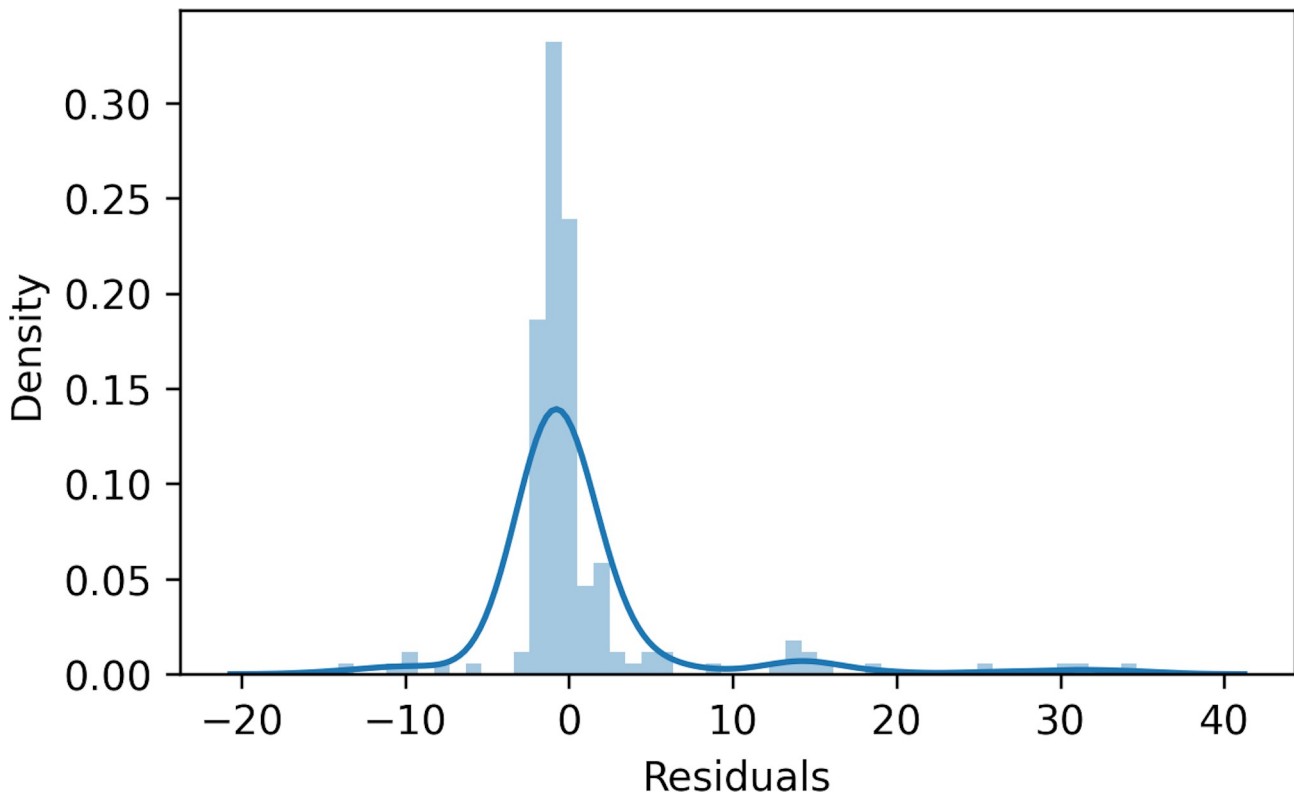

**Fig 5. Residuals of the Decision Tree modeling results.**

Upon analyzing the key features highlighted by both the Decision Tree and Random Forest models, we discovered a consensus regarding the most significant predictors. Figs 6 and 7 illustrate that fault density and average peak ground acceleration (%g) are identified by both models as the top two influential factors in forecasting the number of infrastructure accidents. While considering the frequency of earthquakes by category, the Decision Tree model emphasizes earthquakes classified under EQ-HD2 (magnitude ≥3.5 earthquakes with hypocentral depths less than 20 km) as one of the top three critical features. Conversely, the Random Forest model prioritizes EQ-HD1 (magnitude ≥3.5 earthquakes with hypocentral depths up to 10 km) as a more significant factor. Despite the variation in the specific ranking of these features, the overarching conclusions drawn from both models are in agreement. They show that, in addition to fault density and average PGA, shallow earthquakes of magnitude 3.5 or greater (with hypocentral depths up to 20 km) exhibit a pronounced correlation with the incidence of

**Table 3. Statistical significance tests relative to linear regression model and confidence intervals for RMSE values of different models.**

| Model | Confidence Intervals | t-statistics | p-values |
|---|---|---|---|
| Linear | (14.98, 16.10) | - | - |
| Ridge | (7.44, 8.04) | 25.13 | 7.25e-33 |
| Lasso | (7.44, 8.04) | 25.12 | 7.40e-33 |
| Elastic Net | (7.44, 8.04) | 25.12 | 7.30e-33 |
| Decision Tree | (5.91, 6.43) | 31.01 | 8.45e-38 |
| Random Forest | (5.66, 6.19) | 31.81 | 2.10e-38 |

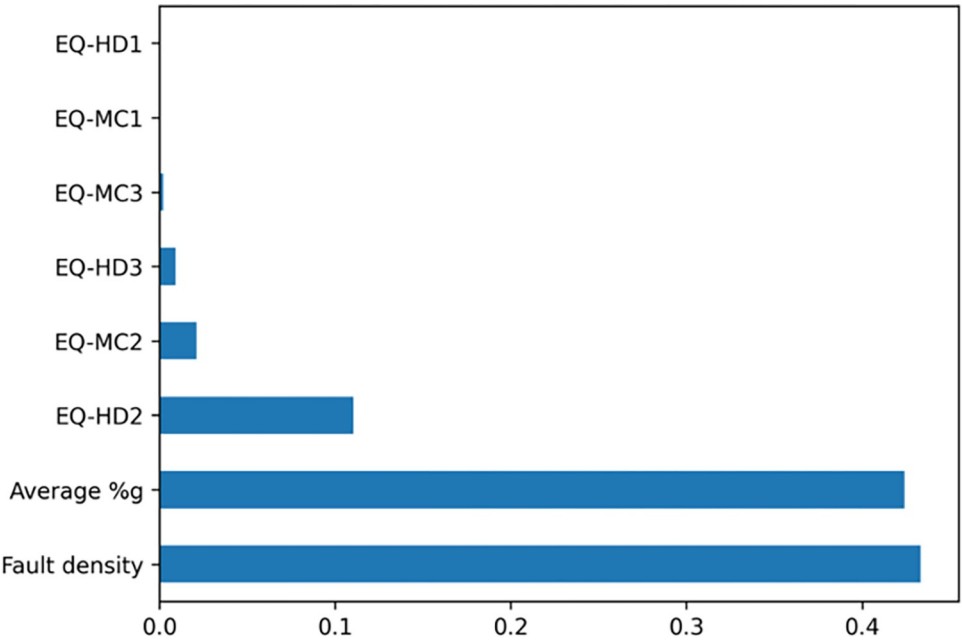

**Fig 6. The Decision Tree model feature importance.**

railway infrastructure accidents. This consistency underscores the critical role these hazard indicators play in influencing infrastructure vulnerability and risk assessment.

Taking into account the outcomes from all models, it becomes evident that non-linear models offer a superior framework for elucidating the complex relationship between railway infrastructure accidents and seismic hazard variables. This observation suggests that the

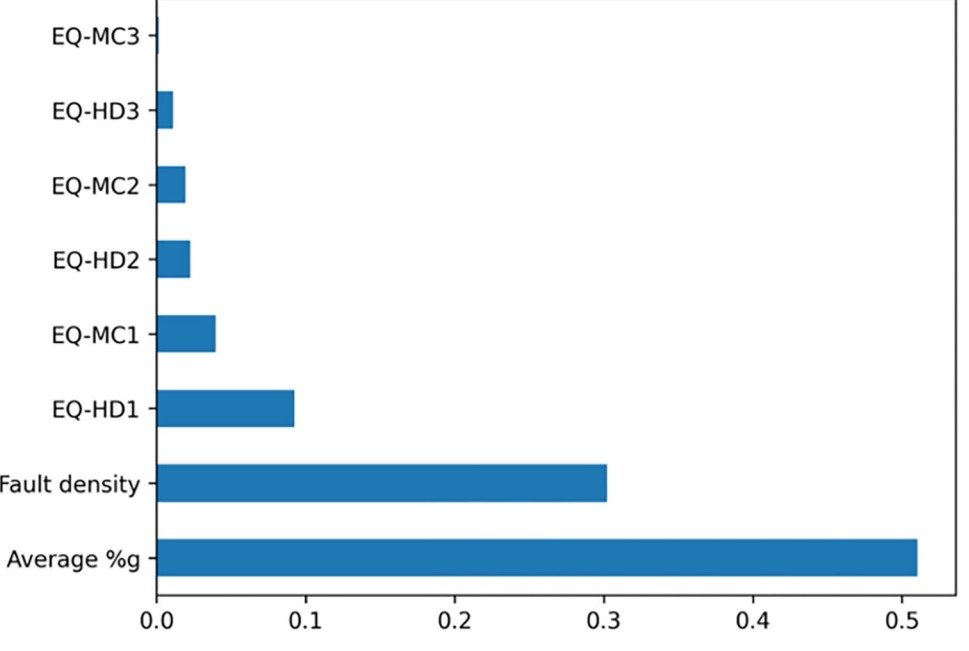

**Fig 7. The Random Forest model feature importance.**

intricacies of this relationship are beyond the explanatory power of simplistic least squares linear regression models. The enhanced performance of ensemble models highlights their ability to capture and interpret the multifaceted and nonlinear interactions inherent in the data, underscoring the necessity for more sophisticated analytical approaches in understanding the dynamics at play in predicting infrastructure vulnerabilities related to seismic and fault activity and ground shaking.

## 4. Modeling discussion

### 4.1 Relationship between railway infrastructure accidents and seismic hazard variables

The Decision Tree model offers insightful predictions regarding the number of infrastructure accidents through its hierarchical structure of decision nodes (Fig 8). The root node, characterized by average %g $\leq$ 0.707, showcases the largest sum of squared residuals, guiding the model's growth through successive parent and child nodes until reaching the predictive leaf nodes (Fig 8). For instance, in a hypothetical scenario within County A, given an average %g > 0.707 and EQ-HD3 $\leq$ 0.002, the model predicts approximately two accidents (with six observations in this category).

Our analysis reveals that an earthquake's impact on railway tracks and infrastructure can vary significantly based on the earthquake's location (depth and distance from the track) and magnitude. Moreover, aseismic events, such as aseismic creep along fault lines—evidenced by displacements without recorded earthquakes (e.g., along the Hayward Fault in Berkeley, Alameda County, California) [4,49]—can also contribute to infrastructure damage. These factors underscore the multifaceted and intricate nature of the relationships under study.

Although our dataset lacks the detailed temporal and spatial granularity that could enhance the precision of our predictions, the Decision Tree's outcomes serve as a valuable preliminary model for forecasting the potential number of railway accidents. With access to more detailed data in the future, we can refine our model and re-evaluate our findings to achieve a more comprehensive understanding of the dynamics influencing railway infrastructure vulnerability to fault and earthquake activity, and ground shaking intensity.

### 4.2 Important features

The fault density and average PGA emerge as the paramount features influencing the ensemble models' predictive accuracy. Railways located in counties with a higher fault density—indicating a greater total length of fault lines—are more prone to damage in the event of nearby

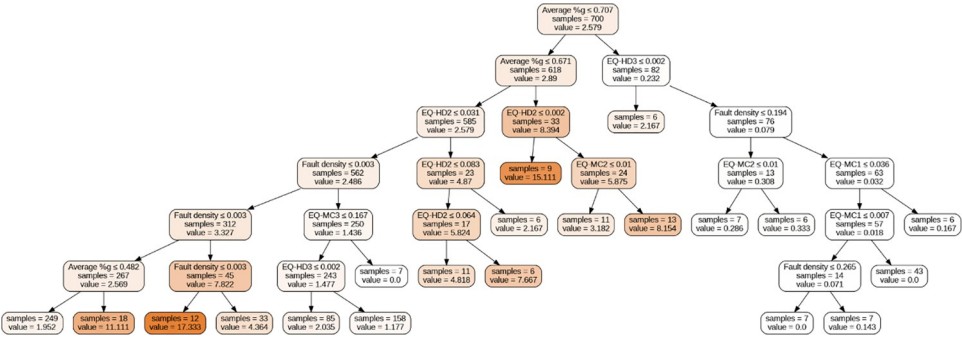

**Fig 8. Tree suggested by Decision Tree model.**

earthquakes. This increased vulnerability can be attributed to the fact that faults represent zones of weakness where seismic or aseismic events are more likely to occur.

Moreover, the average PGA reflects the intensity of ground shaking in a county, which is affected by various factors including the underlying geology (such as soil or rock types), the magnitude and specific characteristics of earthquakes affecting that area, and the proximity to the fault rupture. The considerable variation in fault density and PGA values across West Coast counties underscores these features' effectiveness in identifying areas at higher risk for railway infrastructure accidents and damage (Figs 3 and 4). For instance, counties in western California are characterized by relatively high fault densities and PGA values, which are indicators of seismic hazard. In contrast, counties in Oregon generally have lower fault density and PGA values, suggesting a lower seismic hazard. Consequently, the likelihood of railway infrastructure accidents due to seismic activity is higher in western California than in Oregon.

Additionally, the third most significant feature differs between the Decision Tree and Random Forest models, being EQ-HD2 and EQ-HD1, respectively. Despite this variance, both features relate to the depth of an earthquake's hypocenter, highlighting the depth's critical role in influencing surface infrastructure damage. Earthquakes with magnitude $\geq 3.5$ originating at deeper depths ($>20$ km) tend to have a lesser impact on railway infrastructure compared to those occurring at shallower depths ($\leq 20$ km). Also, the pronounced impact on the non-linear regression model may stem from the abundance of shallow earthquakes—those with hypocentral depths of 20 km or less—primarily observed in California. This characteristic significantly influences the model's performance.

In conclusion, our models indicate that the proximity to fault lines, the severity of ground shaking, and the occurrence of shallow-depth earthquakes significantly affect the frequency of railway infrastructure accidents in West Coast counties. These findings highlight the importance of considering geological and seismic data in assessing and mitigating the risks to railway infrastructure from fault and earthquake activity.

## 4.3 Model limitations

Our study currently grapples with limitations in the granularity of railway infrastructure accident data and the absence of comprehensive information on infrastructure conditions. The available data consists merely of annual records of railway infrastructure accidents, without further detail on the state of railway conditions in each county for each year. Crucially, these records lack specific data on the exact locations of the accidents and the precise timing of the incidents, which could significantly enhance the modeling process. Given the non-linear nature of the relationship between infrastructure accidents and seismic hazard variables, a single earthquake can potentially cause multiple instances of damage across a network, not limited to a single location or piece of infrastructure.

Furthermore, the geographical impact of an earthquake does not respect administrative boundaries; seismic waves can propagate beyond the county's borders where the quake originated, potentially affecting infrastructure in adjacent areas. This cross-border effect depends on several factors, including the earthquake's epicenter location and magnitude. Our current data limitations prevent us from accurately tracing how specific earthquakes contribute to subsequent infrastructure damage.

Looking ahead, should we gain access to more detailed accident data, including precise locations and timings of incidents, our ability to understand and predict the impact of seismic events on railway infrastructure would be greatly enhanced. Our study can also be extended to include lower magnitude earthquakes (M<3.5) and swarm sequences characterized by abundant earthquakes occurring locally due to secondary processes [50]. In the future, with more

comprehensive data, we intend to revisit and refine our models to develop a more accurate predictive framework for assessing the risk and impact of earthquakes on railway infrastructure.

## 5. Financial impact of railway infrastructure accidents on operational plan

Our results suggest that earthquake or fault activity in the western US has affected railway infrastructure; therefore, minimizing exposure of all risks occurring from railway infrastructure should be of concern for increasing passenger safety. However, the financial planning challenges faced by railway operator companies is one of the barriers to increasing safety performance. This issue can happen in both developed and developing countries.

In the financial assessment and sensitivity analysis sections below, we evaluate the damage costs associated with railway infrastructure from 2000 to 2023 to ascertain the financial performance of safety improvement measures.

### 5.1 An analysis of NPV

An evaluation of the Federal Railroad Administration (FRA) data [29], as depicted in Fig 9, provides a comparative analysis of the financial impact on railway infrastructure for California, Oregon, and Washington over a span of 22 years. Utilizing a sensitivity analysis that incorporates the time value of money, with an interest rate benchmarked at 5.5%—representative of the average rate in the United States over a two-decade span—the study reveals that California has incurred the most significant average annual damage costs [51]. These costs aggregate to a total of $240,146,138, equating to an annual mean of approximately $10,441,136. This trend is accentuated by notable economic disruptions, specifically in the years 2003 and 2007. Table 4 presents observations from 2018 to 2022, indicating a pronounced fluctuation in the cost of damages in California, with the figures largely remaining elevated. The damage costs peaked in 2022 at $5,051,303, significantly higher than other years and notably surpassing the 5-year average damage cost of $2,242,674.6.

Contrasting sharply with California, Washington and Oregon typically experience more subdued financial impacts from rail infrastructure damage. However, certain years, most notably 2020, deviate from this trend, with notable spikes in damage costs. In 2020, Washington

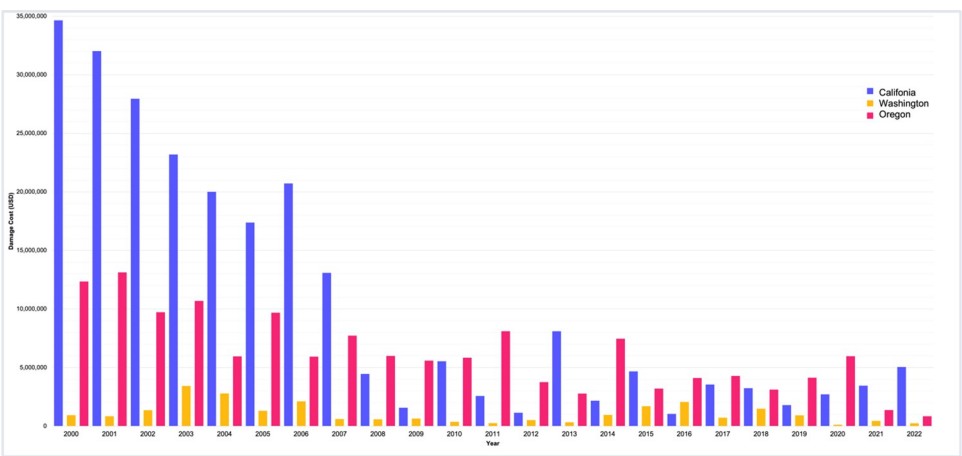

**Fig 9. Comparison of damage cost on rail's infrastructure among California, Oregon, and Washington during the 2000–2022 period (included time-sensitive value).**

**Table 4. The comparison of the damage cost of each state during 2018–2022.**

| Year | Damage cost (unit: USD) | | |
|---|---|---|---|
| | California | Oregon | Washington |
| 2018 | 3,241,804 | 1,490,808 | 3,118,684 |
| 2019 | 1,791,903 | 923,590 | 4,135,514 |
| 2020 | 2,709,152 | 118,119 | 5,969,304 |
| 2021 | 3,452,918 | 441,602 | 1,369,562 |
| 2022 | 5,051,303 | 242,724 | 848,339 |
| Average | 2,242,674.6 | 160,489 | 1,637,441 |

recorded rail infrastructure damages amounting to $5,969,304, a significant increase compared to other years, exceeding the elevated costs observed in California during the same period. This data might suggest the states' vulnerability to specific disruptive events, such as Ferndale earthquakes, that lead to significant infrastructural degradation. From 2018 to 2022, Washington's average annual cost for rail damage is approximately $1,637,441, while Oregon, with the smallest fiscal impact among the three, has an annual average of just $160,489.

The total damage costs in this five year period shows that California has experienced higher financial burdens over other states. The implications of such findings are manifold, warranting a deeper exploration into the causal relationships and potential mechanisms that could elucidate the pronounced disparity in economic vulnerability among these geographically contiguous states.

Nevertheless, when engaging in fiscal analyses related to railway infrastructure, it is imperative to consider the operating distances within each state. Table 5 elucidates the damage cost per mile of operating distance for California, Oregon, and Washington. The data reveals that California incurs a damage cost of $453.25 per operational mile. Oregon, with a significantly lower impact, incurs costs of $67.75 per operational mile. Washington has the highest cost per operational mile at $571.13. These statistics not only manifest substantial disparities in the average damage costs but also in the costs normalized by operating distance across the three states. Such financial metrics are integral to a comprehensive risk assessment and are critical for the strategic allocation of resources aimed at the upkeep of railway infrastructure and the implementation of effective disaster mitigation strategies.

## 5.2 Damage cost sensitivity analysis

Within the framework of this study, a Monte Carlo Simulation (MCS) approach is employed, leveraging a triangular probability distribution to encapsulate the variability of damage costs. This approach provides a probabilistic range for the standard costs associated with various categories of damage: T0 (pertaining to roadbed settlements, washouts, rain, and slide issues), T1 (track irregularities), T2 (rail joint and spike defects), T3 (worn, broken, or defective rail switches), and T4 (flangeway impairments and comprehensive structural defects), all evaluated under normative conditions, as shown in Table 6.

**Table 5. Comparison of 5-years average value and damage cost per operating distance.**

| Value | State | | |
|---|---|---|---|
| | California | Oregon | Washington |
| Operating distance (mile) | 4,948 | 2,369 | 2,867 |
| Damage cost / operating distance (USD/mile) | 453.25 | 67.75 | 571.13 |

**Table 6. Summary of cause codes, causes of accident, and examples.**

| Cause code | Causes of accident | Example |
|---|---|---|
| T0 | Roadbed | Roadbed issues such as settling or softening, as well as damage caused by washouts, rain, slides, floods, snow, or ice are considered additional defects in the track structure. |
| T1 | Track Geometry | Track irregularities, including cross-level irregularities at joints and non-joint locations, as well as irregularities in track alignment such as buckling or sun kinking, along with other track geometry defects, are considered within this category |
| T2 | Rail, Joint Bar, Rail Anchoring | Additional defects related to ways and structures, including misaligned or failed bridges, engineering design or construction issues, clogged flange ways, and faults in the catenary system, are included in this category, with detailed descriptions available in the narrative. |
| T3 | Frogs, Switches, Track Appliances | Additional defects related to ways and structures, including misaligned or failed bridges, engineering design or construction issues, clogged flange ways, and faults in the catenary system, are included in this category, with detailed descriptions available in the narrative. |
| T4 | Other Way and Structure | Additional defects related to ways and structures, including misaligned or failed bridges, engineering design or construction issues, clogged flange ways, and faults in the catenary system, are included in this category, with detailed descriptions available in the narrative. |

The basis for this analysis is a comprehensive data set spanning from 2000 to 2022, incorporating all counties within California, Oregon, and Washington. A Monte Carlo simulation was conducted to enhance the robustness of the sensitivity analysis. The distribution parameters are calibrated with a conservative lower bound set at 10% below the established standard cost, considering factors such as a small number of earthquakes, and an optimistic upper limit set at 60% above the standard cost, accounting for factors such as a large number of earthquakes, high inflation rates, etc. This simulation involved 5,000 iterations to ensure a comprehensive exploration of possible outcomes. The input variable for the simulation was accident cost, including time-sensitive values. This input variable was modeled using a triangular distribution. The sensitivity analysis results derived from this data and the Monte Carlo simulation are presented in Table 7. Furthermore, the study offers a comparative analysis of the deviation in railroad accident costs in the different cause code categories, illustrated in Figs 10–12, which delineates the financial variance attributable to railroad accidents across the tri-state area. This methodology facilitates an in-depth understanding of potential fiscal impacts under a set of scenario conditions, contributing to a robust financial risk assessment in railway infrastructure management.

Moving forward, a sensitivity analysis would provide valuable insights into the fiscal variability revealed in Table 7. Here, the scope will be confined to California to ensure brevity and depth of analysis. This decision is justified by California's high average damage cost values and larger range in deviation from the mode. Focusing on California will allow for a comprehensive yet concise exploration of its unique transportation economics.

The Monte Carlo simulation reveals the economic ramifications of transportation infrastructure deficiencies in California, based oncause codes ranging from T0 to T4. The code T0, delineating issues related to the roadbed, inclusive of environmental detriments, manifests the second lowest economic impact, with a modal cost of $84.06 per mile-year. Conversely, T1, which pertains to complications arising from track geometry, incurs a significantly elevated modal cost of $865.12. T2, indicative of pronounced structural faults, such as bridge misalignments, represents the maximumcost implications, recording a modal figure of $2971.63. T3 encapsulates defects within track components, including switches, culminating in a moderate

**Table 7. Summary of Monte Carlo simulation result by cause code for each state.**

| State | Cause code | Accident Cost (USD/mile-year) | | |
|---|---|---|---|---|
| | | Minimum | Mode | Maximum |
| California | T0 | 93.4 | 84.06 | 149.45 |
| | T1 | 961.24 | 865.12 | 1537.99 |
| | T2 | 3301.81 | 2971.63 | 5282.9 |
| | T3 | 351.39 | 316.25 | 562.22 |
| | T4 | 37.3 | 33.57 | 59.67 |
| Oregon | T0 | 21.48 | 19.33 | 34.36 |
| | T1 | 423.38 | 381.04 | 677.4 |
| | T2 | 376.38 | 338.74 | 602.2 |
| | T3 | 126.18 | 113.56 | 201.89 |
| | T4 | 17.74 | 15.97 | 28.39 |
| Washington | T0 | 726.8 | 654.12 | 1162.88 |
| | T1 | 1876.78 | 1689.1 | 3002.85 |
| | T2 | 1647.41 | 1482.67 | 2635.85 |
| | T3 | 366.7 | 330.03 | 586.73 |
| | T4 | 39.21 | 35.29 | 62.74 |

cost imposition of $316.25. Lastly, T4 encompasses miscellaneous infrastructural and structural concerns, registering the least modal cost of $33.57. This hierarchical codification underpins the strategic allocation of maintenance resources and the prioritization framework for infrastructure refurbishment.

This data-driven analysis through the Monte Carlo method reveals the variable financial risks of infrastructure categories, providing an analytical basis for prioritizing investment and maintenance to mitigate escalating costs.

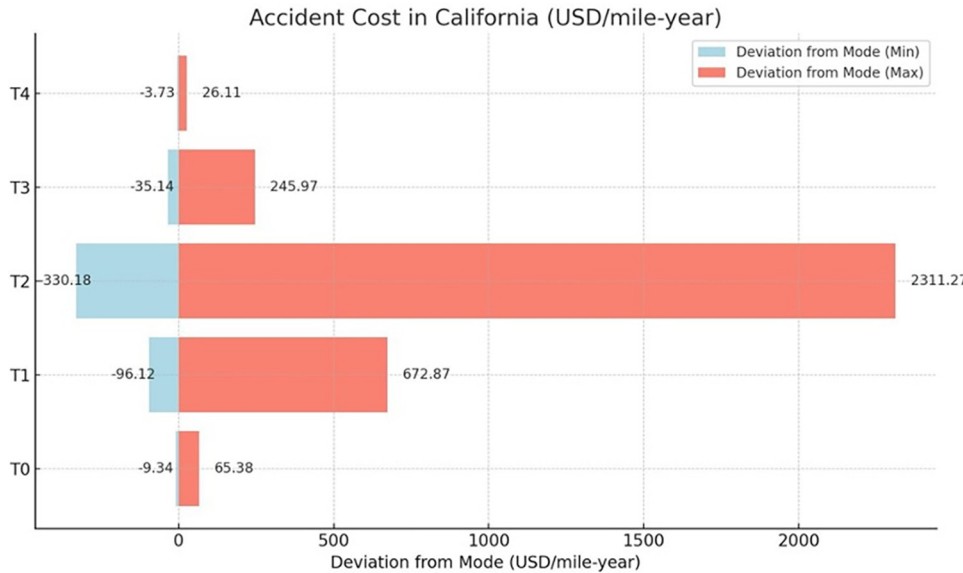

**Fig 10. Comparison of deviation from the mode of railroad accident costs for California in different cause code categories.**

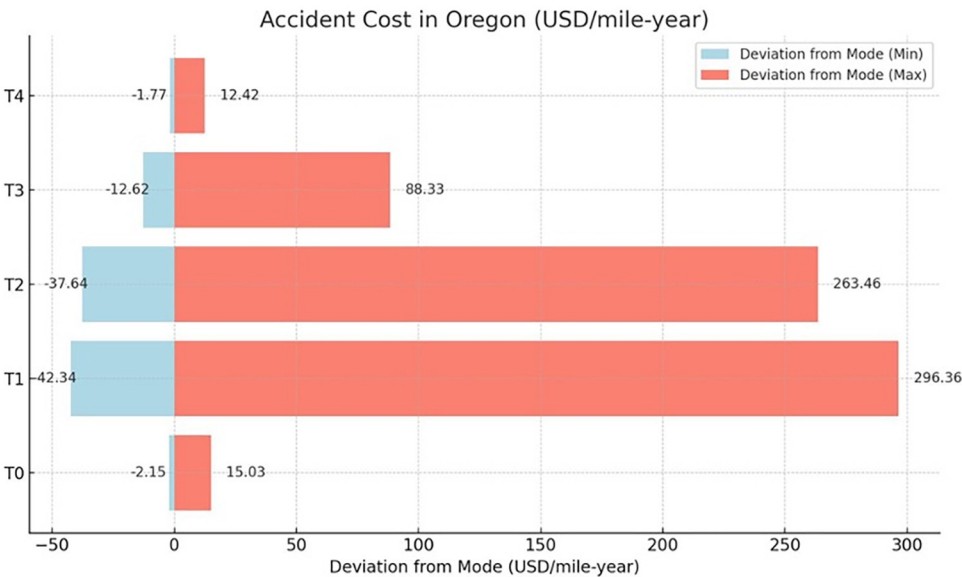

**Fig 11. Comparison of deviation from the mode of railroad accident costs for Oregon in different cause code categories.**

## 6. Recommendations

The development of advanced analytical tools for processing data on seismic hazard in real time to predict potential impacts on railway infrastructure has been a significant area of research and innovation. We recommend tools that combine cutting-edge technologies, including machine learning (ML) and geographic information systems (GIS), to enhance the safety and reliability of railway operations in earthquake-prone areas. One possible tool is combining GIS with real-time seismic data. This could facilitate the visualization and

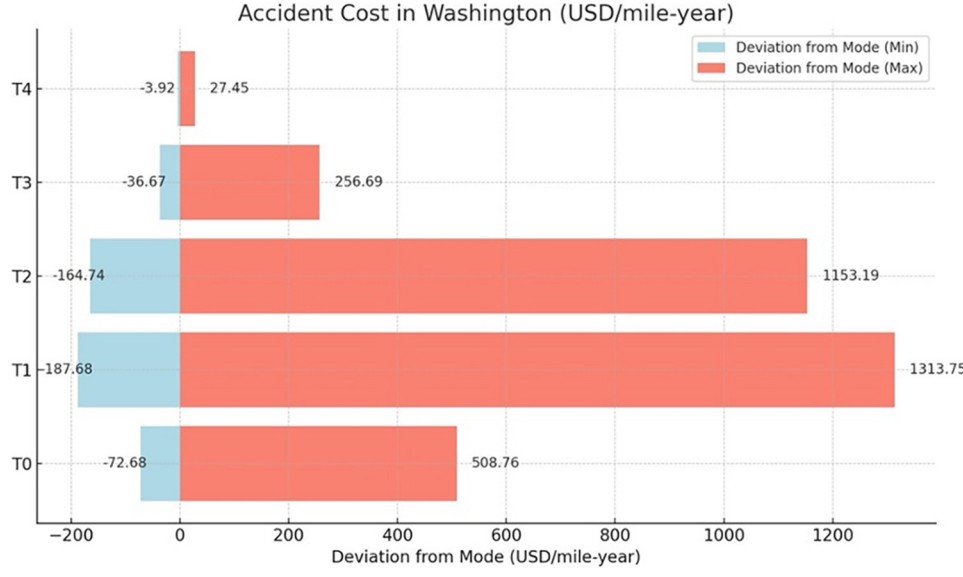

**Fig 12. Comparison of deviation from the mode of railroad accident costs for Washington in different cost code categories.**

examination of how earthquakes affect railway systems. Such tools are capable of outlining both the geographical spread of seismic risks and the susceptibility of railway infrastructure. This type of geospatial analysis aids in determining priorities for emergency response, recovery actions, routing decisions, and strengthening infrastructure. Another tool is Earthquake Early Warning Systems. These systems can employ a network of seismic sensors placed along fault lines and railway routes to identify preliminary tremors (foreshocks) ahead of the main earthquake shocks that may impact key infrastructure. Through real-time analysis of seismic data and earthquake early warning, these systems are capable of forecasting the?likely effects of an earthquake, enabling trains to automatically decelerate or stop. This preventive measure aims to avoid derailments and reduce potential damage.

Effective operational strategies are crucial for enhancing the performance of maintenance activities, particularly regarding railway infrastructure and its vulnerability to earthquake and fault activity, and intense ground shaking. To maintain safety, reliability, and uninterrupted service in areas prone to seismic activity, it is recommended to adopt advanced maintenance, monitoring, and emergency preparedness approaches. Suggested strategies include:

1. Adopting a strict and regular maintenance regimen for all railway infrastructure elements, prioritizing areas most susceptible to damage from seismic events like bridges, tunnels, and rail tracks. This approach should encompass consistent inspections, evaluations of material wear and tear, and structural fortifications.

2. Formulating a comprehensive emergency response plan that incorporates automated train control mechanisms to decelerate or halt trains upon receiving seismic warnings, establishes safe zones for trains during earthquakes, and assembles quick-reaction teams for immediate inspection and restoration after seismic events.

3. Organizing frequent training and simulation exercises for both maintenance and emergency response personnel to guarantee their preparedness. Training should cover the latest in maintenance technologies and practices, alongside emergency procedures.

4. Collaborating with local communities, governmental bodies, and other relevant stakeholders to unify efforts and disseminate information about railway safety and emergency measures. Part of this effort includes educating the public on how to prepare for and react to earthquakes.

Implementing these strategies will bolster the resilience of railway infrastructure against earthquakes, improving both the safety of passengers and the continuity of railway services.

## 7. Conclusion

The relationship between railway infrastructure accidents and seismic hazard variables is not explained by simple linear models, underscoring the multifaceted nature of earthquake impacts. This complexity may be partly due to the fact that the occurrence of railway infrastructure incidents in the aftermath of an earthquake is not limited to a single event or location but can manifest as multiple incidents across various locales.,Non-linear models outperform linear models in their predictive capabilities and shed light on the crucial determinants of railway infrastructure vulnerabilities along the U.S. West Coast. These models pinpoint fault density, ground shaking intensity, and the prevalence of shallow earthquakes (with hypocentral depths of less than 20 km and magnitudes of 3.5 or greater) as pivotal factors that escalate the risk of railway infrastructure accidents. Such insights call for a strategic approach to mitigating the seismic risks faced by railway systems, emphasizing the need for advanced modeling

techniques to accurately predict and prepare for the seismic challenges inherent in these tectonically active regions.

Our financial evaluation, underpinned by Net Present Value and Monte Carlo Simulation, provides a concise synthesis of the potential economic fallout from unanticipated disruptions due to fault and earthquake activity on railway infrastructure. Overall California's five-year data reflects an average damage cost of USD 2,242,674.6 per mile, indicative of the severity of the financial consequences. This study further quantifies the fiscal impact of railway infrastructure damage costs in five cause categories with substantial variations revealed across the different categories. This figure, coupled with thenon-linear relationship between seismic hazard variables and railway infrastructure accidents, underscores the pressing need for a combined fiscal and geospatial strategy in railway risk management.

## Supporting information

**S1 File. Earthquakes M3.5 or above in CA, WA, and OR.** This dataset compiles earthquake occurrences of magnitude 3.5 or greater from 2000 to 2022 in the states of California, Washington, and Oregon. It includes details such as the date, time, location, magnitude, and depth of each earthquake. The data is sourced from the United States Geological Survey (USGS) to assist in seismic research and analysis.
(CSV)

**S2 File. Data availability.** A comprehensive dataset detailing earthquakes in California, Oregon, and Washington from 2000 to 2022 with magnitudes of 3.5 or greater. It includes annual statistics such as the number of earthquakes per state, minimum, mean, and maximum depths, and magnitudes recorded. This data supports the study on seismic hazards and their impact on railway infrastructure along the U.S. West Coast. Data sources include the U.S. Geological Survey (USGS).
(PDF)

## Acknowledgments

Maps were created using the QGIS software.

## Author Contributions

**Conceptualization:** Patcharaporn Maneerat, Panrawee Rungskunroch.

**Data curation:** Panrawee Rungskunroch.

**Formal analysis:** Patcharaporn Maneerat.

**Funding acquisition:** Panrawee Rungskunroch.

**Methodology:** Patcharaporn Maneerat.

**Project administration:** Panrawee Rungskunroch.

**Software:** Patcharaporn Maneerat.

**Supervision:** Patricia Persaud.

**Validation:** Patricia Persaud.

**Writing – original draft:** Patcharaporn Maneerat, Patricia Persaud.

**Writing – review & editing:** Panrawee Rungskunroch, Patricia Persaud.

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
