## [Decision Letter · Decision Letter 0]

10 Jun 2024

PONE-D-24-16061Seismic Hazard Analysis and Financial Assessment of Railway Infrastructure: A Machine Learning ApproachPLOS ONE

Dear Dr. Rungskunroch,

Thank you for submitting your manuscript to PLOS ONE. After careful consideration, we feel that it has merit but does not fully meet PLOS ONE’s publication criteria as it currently stands. Therefore, we invite you to submit a revised version of the manuscript that addresses the points raised during the review process.

**Please consider all comments **

We look forward to receiving your revised manuscript.

Kind regards,

Ahmed Mancy Mosa, Ph.D.

Academic Editor

PLOS ONE

Journal Requirements:

Rajamangala University of Technology's Postdoctoral and Postgraduate Talent Resource Development to encourage in deep research for improving competitiveness of industry contract code [B13F660068]

The first and second authors sincerely appreciate the financial assistance provided by the NSRF through the Rajamangala University of Technology's Postdoctoral and Postgraduate Talent Resource Development to encourage in deep research for improving competitiveness of industry contract code [B13F660068]. The second authors also thanks to the RISEN funding for one year at the University of California, Berkeley.

Rajamangala University of Technology's Postdoctoral and Postgraduate Talent Resource Development to encourage in deep research for improving competitiveness of industry contract code [B13F660068]

5. Thank you for uploading your study's underlying data set. Unfortunately, the repository you have noted in your Data Availability statement does not qualify as an acceptable data repository according to PLOS's standards.

7. We note that Figures 1, 2, 3 and 4 in your submission contain map images which may be copyrighted. All PLOS content is published under the Creative Commons Attribution License (CC BY 4.0), which means that the manuscript, images, and Supporting Information files will be freely available online, and any third party is permitted to access, download, copy, distribute, and use these materials in any way, even commercially, with proper attribution. For these reasons, we cannot publish previously copyrighted maps or satellite images created using proprietary data, such as Google software (Google Maps, Street View, and Earth). For more information, see our copyright guidelines: http://journals.plos.org/plosone/s/licenses-and-copyright.

We require you to either present written permission from the copyright holder to publish these figures specifically under the CC BY 4.0 license, or remove the figures from your submission:

a. You may seek permission from the original copyright holder of Figures 1, 2, 3 and 4 to publish the content specifically under the CC BY 4.0 license.  

8. Please remove your figures from within your manuscript file, leaving only the individual TIFF/EPS image files, uploaded separately. These will be automatically included in the reviewers’ PDF.

Reviewers' comments:

Reviewer's Responses to Questions

**Comments to the Author**

1. Is the manuscript technically sound, and do the data support the conclusions?

Reviewer #1: Partly

Reviewer #2: Partly

2. Has the statistical analysis been performed appropriately and rigorously? 

Reviewer #1: I Don't Know

Reviewer #2: No

3. Have the authors made all data underlying the findings in their manuscript fully available?

Reviewer #1: No

Reviewer #2: Yes

4. Is the manuscript presented in an intelligible fashion and written in standard English?

Reviewer #1: Yes

Reviewer #2: Yes

5. Review Comments to the Author

**Reviewer #1:** The authors of the paper touched upon a very important topic. On the one hand, they determined the relationship between the frequency of earthquakes and railroad accidents, and on the other hand, they linked these data to the costs of railroad infrastructure.

However, the methods used by the authors can be called classical. It would be interesting to use modern machine learning methods. It is unclear why the authors, for example, did not use deep neural networks. In my opinion, for publication in a journal of this level, the authors should have conducted a deeper analysis in at least one of the two selected areas.

Comments

1. The manuscript can be divided into two parts. The first part, devoted to establishing the connection between earthquakes and railroad accidents, is technically competent. Classical methods are used, the literacy of which is not in doubt. As for the second part - proposals for the development of railroad infrastructure - I cannot give an assessment.

2. I can't spin off the dataset, so I can't check the validity of the statistical analysis.

3. The authors did not provide full access to all data underlying the results of their manuscript.

4. The manuscript is written in clear language.

**Reviewer #2:** The manuscript should provide more detailed descriptions of the datasets used. While it mentions the sources (Federal Railroad database and U.S. Geological Survey database), it should include details such as the number of data points from each, time periods covered, and any preprocessing steps taken to refine the inputs to the model.

In terms of experiment set-up, the methodology section should elaborate on control measures - for example, if there were any control variables and/or assumptions made during model training/testing, these should be explicitly stated. Further, while the document captures the split of data into training and validation sets, explaining the approach used (e.g., proportions, randomization methods) would drive transparency. Further, information number of repeated experiments to arrive at the results is also missing. For example, specifying cross-validation techniques/detailing the number of folds will improve the result reliability.

While the manuscript compares different models, it should also provide statistical significance of tests across these models. Including confidence intervals or related metrics will help here.

The details around monte-carlo simulation are also missing - detailing the number of simulations and the distribution of input variables is needed.

Clarification is needed on metrics used for performance evaluation and their suitability for this research - for instance the rationale behind using R-squared, Mean Absolute Error (MAE), and Root Mean Squared Error (RMSE) for model evaluation.

Other general suggestions:

Please provide the definition or citation to technical terms - for instance peak ground acceleration is not defined when it is used first. Similarly, there is lack of consistency in usage of technical terms -seismic variables and earthquake variables are used interchangeably which can create confusion to the reader

6. PLOS authors have the option to publish the peer review history of their article (what does this mean?). If published, this will include your full peer review and any attached files.

Reviewer #1: **Yes: **Alexey Osipov

Reviewer #2: **Yes: **Vineet Gupta

---

## [Author Response · Author response to Decision Letter 0]

29 Jun 2024

Response to reviewers file is enclosed in the system. All changes in the revision version are highlighted in navy blue.

Here's authors' response to the editor's comments. 

1. We re-uploaded the revision version of manuscript in .docx format following your guidelines.

2. Yes, We revised our cover letter and put the sentences "The funders had no role in study design, data collection and analysis, decision to publish, or preparation of the manuscript". Please re-check the document.

3. Regarding the Figures 1-4, we got the confirmation from editor that those are able to proceed. Figure 1-4 are created by using QGIS, which is a free and open-source geographical information system.

---

## [Decision Letter · Decision Letter 1]

22 Jul 2024

Seismic Hazard Analysis and Financial Impact Assessment of Railway Infrastructure in the US West Coast: A Machine Learning Approach

PONE-D-24-16061R1

Dear Dr.  Rungskunroch,

We’re pleased to inform you that your manuscript has been judged scientifically suitable for publication and will be formally accepted for publication once it meets all outstanding technical requirements.

Kind regards,

Ahmed Mancy Mosa, Ph.D.

Academic Editor

PLOS ONE

Additional Editor Comments (optional):

Reviewers' comments:

Reviewer's Responses to Questions

**Comments to the Author**

1. If the authors have adequately addressed your comments raised in a previous round of review and you feel that this manuscript is now acceptable for publication, you may indicate that here to bypass the “Comments to the Author” section, enter your conflict of interest statement in the “Confidential to Editor” section, and submit your "Accept" recommendation.

Reviewer #2: All comments have been addressed

2. Is the manuscript technically sound, and do the data support the conclusions?

Reviewer #2: Yes

3. Has the statistical analysis been performed appropriately and rigorously? 

Reviewer #2: Yes

4. Have the authors made all data underlying the findings in their manuscript fully available?

Reviewer #2: Yes

5. Is the manuscript presented in an intelligible fashion and written in standard English?

Reviewer #2: Yes

6. Review Comments to the Author

Reviewer #2: The authors have addressed the feedback provided in the previous review and have captured the responses well in the addendum to the submission.

7. PLOS authors have the option to publish the peer review history of their article (what does this mean?). If published, this will include your full peer review and any attached files.

Reviewer #2: **Yes: **Vineet Gupta

---

## [Editor Report · Acceptance letter]

3 Aug 2024

PONE-D-24-16061R1 

PLOS ONE

Dear Dr. Rungskunroch, 

I'm pleased to inform you that your manuscript has been deemed suitable for publication in PLOS ONE. Congratulations! Your manuscript is now being handed over to our production team.

Kind regards, 

on behalf of

Dr. Ahmed Mancy Mosa 

Academic Editor

PLOS ONE